# A Robust Early Warning System for Preventing Flash Floods in Mountainous Area in Vietnam

**Thanh Van Hoang** [1,*] **, Tien Yin Chou** [1] **, Ngoc Thach Nguyen** [2] **, Yao Min Fang** [1] **, Mei Ling Yeh** [1] **, Quoc Huy Nguyen** [3] **and Xuan Linh Nguyen** [3]

[1] Geographic Information Systems Research Center, Feng Chia University, Taichung 40724, Taiwan; jimmy@gis.tw (T.Y.C.); frankfang@gis.tw (Y.M.F.); milly@gis.tw (M.L.Y.)

[2] VNU, University of Science, Faculty of Geography, No 334 Nguyen Trai, Thanh Xuan, Hanoi City 100000, Vietnam; nguyenngocthachhus@vnu.edu.vn

[3] PhD Program of Civil and Hydraulic Engineering, College of Construction and Development, Feng Chia University, Taichung 40724, Taiwan; st_huy@gis.tw (Q.H.N.); st_linh@gis.tw (X.L.N.)

[*] Correspondence: van@gis.tw; Tel.: +886-42451-6669

**Abstract:** The early-warning model for flash floods is based on a hydrological and geomorphological concept connected to the river basin, with the principle that flash floods will only occur where there is a high potential risk and when rainfall exceeds the threshold. In the model used to build flash-floods risk maps, the parameters of the basin are analyzed and evaluated and the weight is determined using Thomas Saaty's analytic hierarchy process (AHP). The flash-floods early-warning software is built using open source programming tools. With the spatial module and online processing, a predicted precipitation of one to six days in advance for iMETOS (AgriMedia—Vietnam) automatic meteorological stations is interpolated and then processed with the potential risk maps (iMETOS is a weather-environment monitoring system comprising a wide range of equipment and an online platform and can be used in various fields such as agriculture, tourism and services). The results determine the locations of flash floods at several risk levels corresponding to the predicted rainfall values at the meteorological stations. The system was constructed and applied to flash floods disaster early warning for Thuan Chau in Son La province when the rainfall exceeded the 150 mm/d threshold. The system initially supported positive decision-making to prevent and minimize damage caused by flash floods.

**Keywords:** early warning; flood; flash floods; analytic hierarchy process; threshold; disaster management

## 1. Introduction

The weather in Vietnam in recent years has been increasingly unusual. Droughts, floods, landslides, thunderstorms and other storms have complicated developments, seriously affecting the country's economy, which depends heavily on agricultural production. In particular, Vietnam is considered to be one of the countries most severely affected by climate change due to its long coastline. If sea level rises by 1 m, 40% of the Mekong Delta area and 10% of the Red River Delta area will be flooded, directly affecting 20–30 million people.

### 1.1. Unusual Weather throughout the Country

In 2016 and early 2017, anomalous, increasingly intense weather occurred throughout the country. Specifically, in the dry season in 2016, many places in South and Central Vietnam were dry due to a rainfall shortage of 30–40% and the flow of small rivers led to saline intrusion one month earlier in these regions. In the river mouth of Central Vietnam and especially in the Mekong Delta, in many

places where saline intrusion is 80–100 km or more, farmers face problems because of the salinity limit and the lack of fresh water for living and production is very serious.

In Central Vietnam, the flood and rain came late yet caused flooding; the floods lasted many days in the last months of 2016, causing great damage to property and people. In the north, the first cold spell came earlier than usual; however, people rarely felt the cold air of the winter, because alternating cold spells had days of high temperature.

Unseasonal rainfall damages the production of winter–spring crops as well as fruit trees. According to meteorological forecasting experts, there are many causes, most of which are due to climate change having changed certain natural laws. The meteorological forecaster, Ms Le Ha (Medium and Long-term Meteorological Forecasting Division, National Center for Hydro-Meteorological Forecasting, leha246@gmail.com) said that the weather is currently neutral and tends to move towards El Niño (usually associated with drought), so the rainy season in South Vietnam has come earlier than normal for many years.

Every year, there are about 10 to 15 flash floods in the northern, central, highland and southeast regions of Vietnam. Due to extreme weather, heavy and prolonged rain is the cause of flash floods and landslides, sweeping trees, rocks and people and property where it passes.

Therefore, natural disasters combined with human factors are causing more floods, flash floods and landslides. If this situation cannot be changed and flooding and landslides continue, outcomes will not be improved.

*1.2. Purpose of this Work and its Background*

In recent years, floods have "surrounded" the Vietnamese provinces of Yen Bai, Son La, Lao Cai, Phu Tho, Hoa Binh, Thanh Hoa, Nghe An and Ha Tinh. As of 7:00 A.M. on 24 July 2018, the number of people killed by flooding was 27, with Son La Province having the highest loss, with 13 people dead. Additionally, rain and flooding also cause damage to property and transport infrastructure in many localities.

Therefore, in addition to urgent measures to overcome immediate consequences, such as support for money, food, medicine and construction materials for people in need, the construction of an early-warning system to mitigate flash flooding is necessary for more sustainable and proactive natural disaster prevention and control measures.

In the Digital Age, many countries have established disaster early-warning systems [1–9]. However, this study is the first research in Vietnam to integrate rainfall data with statistical weather data, satellite image analysis, an MCA (Multi Criteria Model) model, webGIS, Flash Flood Potential Index (FFPI) and iMETOS data (auto-rainfall station) to analyze and provide flash-flood risk information to residents.

Different from a conventional flood, a flash flood refers to a flood with high velocity, containing many debris and occurring unexpectedly in minor basins with sloping terrain, thus resulting in massive destruction. The development of flash floods is closely linked to the intensity of rainfall, climate conditions, topographic features, human activities and the flood drainage conditions of basins. Flash floods normally appear only a few hours after heavy rain exceeds a threshold level [10]. Unfortunately, the data on flash floods are usually in short supply and not systematic; thus, it is hard to use common methods to make hydrological predictions for flash-flood forecasts or warnings [10].

The primary theoretical basis of this research is the basin approach [11]. In this approach, the basin is considered to be a relatively closed system consisting of small tributaries, and, when it rains, the parameters of the buffer surface will determine the mode of flow movement and accumulation in basin boundaries. In mountainous areas with large basin slopes, when rainfall exceeds the threshold, the surface flow will accumulate to form a flash flood. Each basin produces a different mechanism of flash-flood formulation.

The map of flash flood risks shows the likelihood of a flash flood occurring during a minimum period of one calendar year, usually 20–40 years [12,13]. Therefore, depending on the different rainfall distribution by space, flash floods can appear at different times in many different locations and may

be repeated over time [10,14,15]. In the dry season in 2016–2017, South Vietnam, including Ho Chi Minh City, there was some unseasonal and heavy rainfall; the number of days of rain and the total rainfall of the months in the dry season exceeded the average of many years in the same period [16]. Today, the formulation of the map of flash flood risks still has various approaches and methods and produces different results. There are many hydrological-hydraulics methods (or models) applied, such as the North American Mesoscale Model (NAM), the River Analysis System (RAS) developed by the Hydrologic Engineering Center (HEC) in Davis, California (HEC-RAS) or the Soil and Water Assessment Tool (SWAT) [11,12]. Most maps of flash floods are developed based on the side flash flood approach [13]. Therefore, the maps developed largely project along the sloping sides, with a lack of detailed features; thus, application is limited.

In this research, the basin parameters were analyzed and processed using the multi-criteria analysis (MCA) model, through which the entire basin area was evaluated by pixel and the results map is described in detail, meeting the requirements of the district-level research scale (1:10,000) and commune-level scale (1:5000).

## 2. Materials Research

### 2.1. Study Site

A mountainous district in the northwest of Son La province (3/4 is high mountain—Figure 1), Thuan Chau is located along National Highway 6, has a natural area of 154,126 ha and is inhabited by several ethnic minorities (Thai: 74.05%; H'Mong: 11.16%; Kinh: 9.32%; Khang: 2.57%; and other ethnic groups: 2.94%). According to the Statistical Yearbook of Viet Nam 2016 on the website of General Statistics Office, Vietnam, Son La's population in 2016 was around 1,259,026, whereas 153,000 people living in Thuan Chau (study site) are affected by flooding [17].

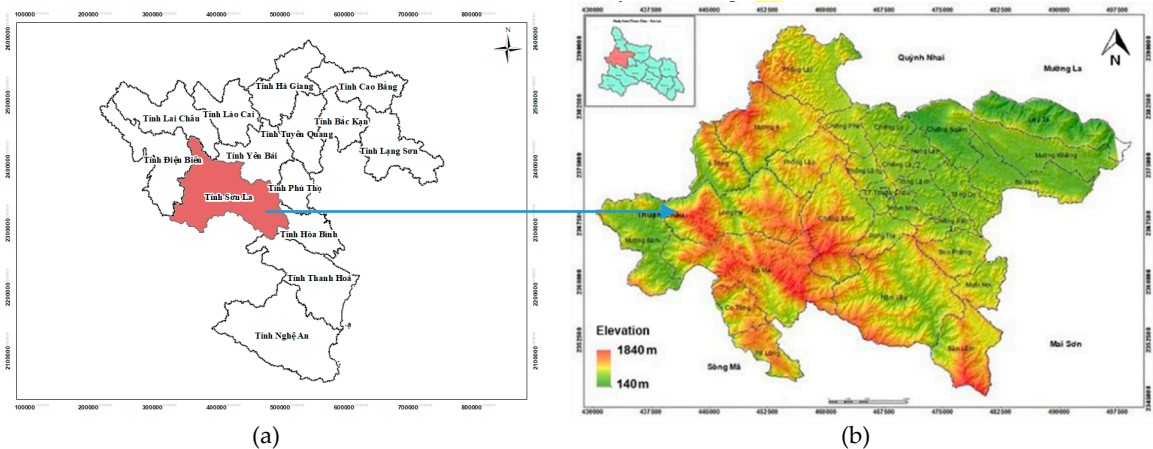

(a)     (b)

**Figure 1.** (**a**) Position of Thuan Chau district and (**b**) digital elevation model (DEM).

Thuan Chau's terrain is elevated, sloping and clearly divided: the highest point over sea level is the Copia peak (1817 m) and the lowest point is Song Da (200 m). In the rainy season, Thuan Chau suffers a lot of natural disasters, such as landslides or flash floods. Over the past few years, for many reasons (including climate change and deforestation), flash floods have started to grow in terms of intensity and frequency, causing severe damage to local communities. As such, research on the development of an early-warning system for flash floods at district level has become an imperative, urgent and practical requirement. With this system, information alerts can be transmitted to different people in various ways, such as message boards, SMS and web pages or can be converted to traditional warning signals (speakers, gongs). Accordingly, local people and managers can make appropriate decisions to prevent natural disasters.

In river basins, to develop a map of flash-flood and mud-flood risks, factors such as landslide, maximum rainfall, the cumulative value of surface topography, surface characteristics, soil characteristics, the weathered shell of the surface and the average slope of tributaries are included as the input data for the analytical model, constructed with a detailed level of research [12,18].

## 2.2. The Theoretical Model in Flash Flood Warning

The general principle of the model is that flash floods will only occur in locations with high potential risks and when rainfall exceeds the flood level. This concept is illustrated in Figure 2.

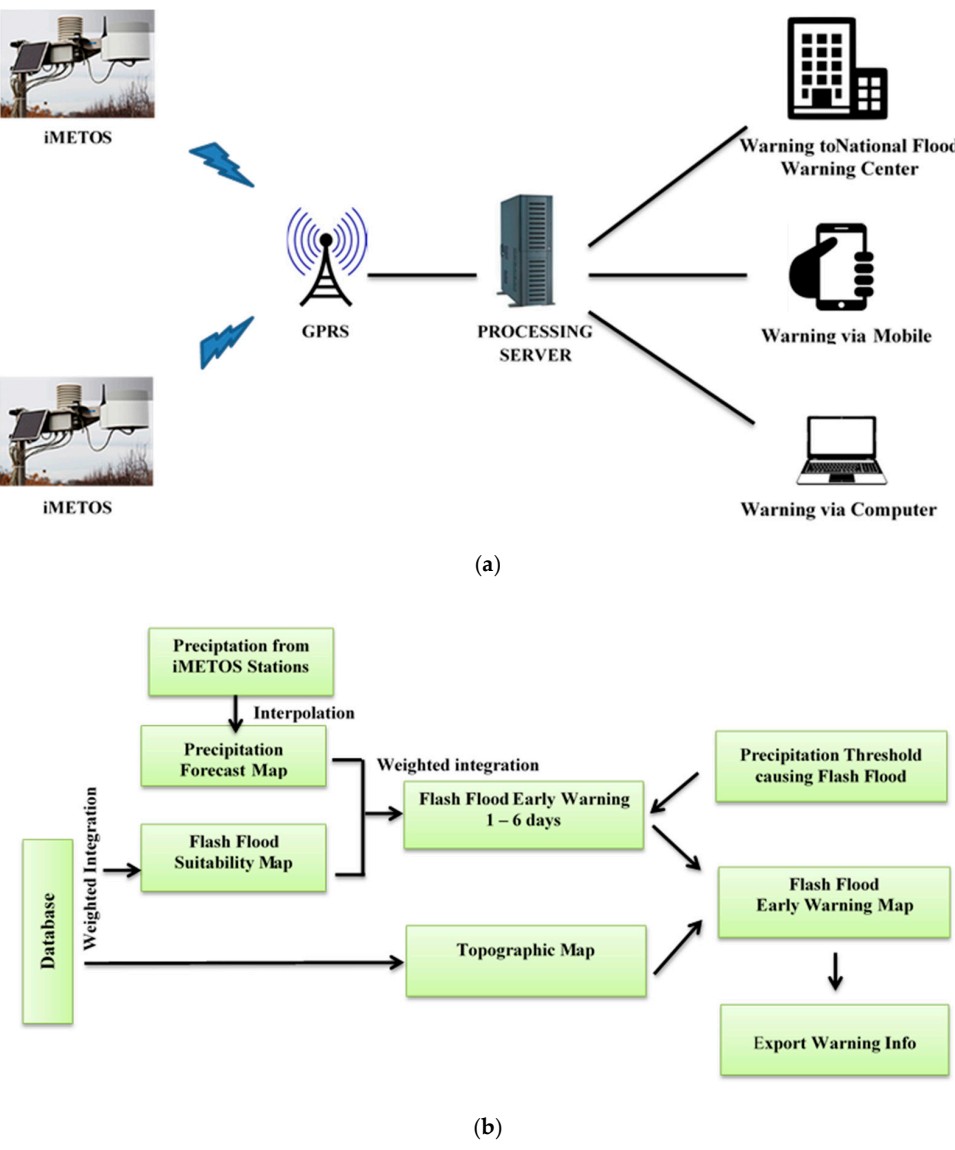

**Figure 2.** (**a**) Model of information processing and integration, (**b**) Workflow of the processing server for early flash flood warning.

As such, in order to obtain an early warning of flash-flood risks, work needs to be done, including: (1) the development of a map of potential flash-flood risks; (2) the development of the model on flash-flood warnings; (3) the development of an iMETOS automatic meteorological station system. These stations should have a 10–15 km active radius, be directly connected to the global meteorological network (www.meteoblue.com) and receive any information about meteorological conditions in the past 30 days, the current weather and weather forecasts for 1–6 days for the station location. It is a

dedicated climate station system with many new features [8,16] approved by the Ministry of Natural Resources and Environment to construct under the Law on Meteorology and Hydrology 2016 [19]; (4) the development of online WebGIS software operating in an internet environment to quickly process rainfall forecasts and integrate them into the potential flash flood risks. Where rainfall exceeds the threshold level, it is possible to quickly identify and provide timely information on the expected time of flash-flood generation and the locations where flash floods can occur in varying degrees.

## 3. Research Methodology

### 3.1. Data Used in the Research

Depending on the research map scale, a number of information layers reviewed will be different. For the 1:10,000 scale for the district level, the information layers included in the evaluation are: mean annual rainfall of several years (RM), topographic wetness index (TWI), average slope of tributes (SB), landslide density (LS), geomorphology (GM), soil (S), forest (FR) and river density (RD). The evaluation rating is divided into five levels: Level 1: very low; Level 2: low; Level 3: medium; Level 4: high; and Level 5: very high [20].

The ratings for each information layer are determined by the expert coefficient and pairwise comparative analysis (analytic hierarchy process, AHP) [21].

There are nine types of map to be collected from difference sources. This research collected not only statistical data but also real-time information for rainfall amount from the auto-rainfall stations (iMETOS). Statistical data were collected over 33 years and included daily rainfall amount and daily temperature data. Most of these data were collected from the Ministry of Agriculture and Development and the Vietnam Ministry of Natural Resources and Environment, as shown in Table 1.

**Table 1.** Materials used in the research.

| No. | Type of Material | Features | Source of Data |
|-----|------------------|----------|----------------|
| 1 | Topographic map | Scale 1: 10,000 | Ministry of Natural Resources and Environment (MONRE) |
| 2 | Forest map | Scale 1: 10,000 | Institute of Forest Planning and Design (Analysis from SPOT (Satellite for observation of Earth) and field survey) |
| 3 | Covering layer map | Scale 1: 10,000 | Analysis from Landsat 8; OLI (Operational Land Imager) image |
| 4 | Soil map | Scale 1: 10,000 | Institute of Soils and Fertilizers, Ministry of Agriculture and Rural Development |
| 5 | Geomorphological map | Scale 1: 10,000 | Formed according to the original morphology |
| 6 | Landslide risk map | Scale 1: 10,000 | Formed using GIS (Geographic Information System) |
| 7 | Historical rainfall data | 52 years (from 1960 to 2015) | Institute of Meteorology, Hydrology and Climate Change (Ministry of Natural Resources and Environment) |
| 8 | Rainfall forecast | Data | iMETOS station data of the research |
| 9 | Historical flood data from 2000 to 2016 | Position, level | Statistics and field interviews |

### 3.2. Muti-Criteria Analysis Model

This model integrates the hydrological and basin geomorphology models assisted by GIS technology [20]. Applying this method, the research focuses on identifying the factors that cause a flash flood in the basin, including soil properties, vegetation cover, basin slope, river density and cumulative flow, and, in comparison with statistics, is used to classify the potentiality of flash-flood generation of

each information layer. Geographic information system is used for weighing and integrating spatial elements with the ratings and flow at each pixel of the raster-form map. This method is qualitative and subjective in nature but is suitable for the conditions of small and medium-sized river basins, where there is a lack of meteorological and hydrological stations [20].

### 3.3. Formulation of Space Model in Early Flash Flood Warning

The threshold for flash-flood generation in serving early warnings is very important; however, it is hard to determine in an accurate manner [2,3,13,14]. Thresholds of flash-flood generation vary widely, ranging from 100 mm/h to 220 mm/d, depending on the basin surface. Most flash-flood basins have an average slope of more than 30% and forest cover of less than 10% of the basin's surface [13]. Researchers of flash floods in Vietnam have divided the probability of flash floods into four levels [22]: Level I (very high risk of flash flood), in basins where all four natural condition factors occur at the same time (relatively frequent rainfall of at least 100 mm/d, average slope of more than 30%, forest coverage of less than 10%, medium and low permeability); Level II (high risk of flash flood), in basins where three natural condition factors occur at the same time (relatively constant rainfall of at least 100 mm/d, with two of the three remaining factors of Level I); Level III (medium risk), in basins with relatively constant rainfall of at least 100 mm/d and one of the remaining three factors; and Level IV (less risk of flash flood), none of the indicators mentioned above. Because flash floods normally appear very quickly and happen in a short time, the flash-flood forecast, or warning, must also be made quickly and some flash floods are even completely unpredictable. The method of the threshold of flash-flood generation facilitates the identification of regions having risks of flash flood and partial rains.

The function for the calculation of early flash flood warning is (Equation (1)):

$$Fr = \sqrt{f.p},$$ (1)

where *Fr* is the early warning map of flash-flood risks; f is the maximum rainfall forecast; and p is the potential flood risk map (FFPI is the flash flood potential index).

## 4. Results

### 4.1. Formulation of the Thuan Chau Flash Flood Risk Map

In reference to some published research [2,15], it is possible to evaluate the layers of basin surface associated with flash-flood risks in the following tables (Tables 2 and 3).

**Table 2.** Flash flood risk evaluation for geomorphology (GM), soil (S) and forest (FR) information layers.

| Evaluation Level | FR | GM | S |
|---|---|---|---|
| Level 1 | Bamboo mixed forest | Surface height is over 1,000 m a.s.l; curved; weak synthetic abrasion<br>Top surface is below 1,000 m a.s.l; arched form; surface sliding | Rocky mountain |
| Level 2 | Medium broad-leaved evergreen forest<br>Bamboo forest | Sloping ramps on limestone; sloping, very sloping; wash drift, landslide<br>Sloping, landslide washed on limestone, lime alternating; very sloping, up to straight sloping, foot slope more relaxed; wash drift, landslide | Reddish brown soil on limestone<br>Reddish brown soil on basic and neutral magma |
| Level 3 | Evergreen broad-leaved forest<br>Evergreen restored broad-leaved forest<br>Forest on rocky surface mountains<br>Plantation forest | Slope slides are quite strong; slightly concave or crooked, top slope 25–30°, under 30–40°; landslide on the original stope slides<br>Ribs worn on different rocks; slightly convex, sloping average of 15–25°, double place 20–25°; synthetic abrasive<br>Medium slope; fairly straight or slightly concave, with average slope of 25–30°; landslide on thick weathering crust, sliding on original rock | Gold soil on sandstone |
| Level 4 | Residence<br>Other land | Surface of flood accumulation; ribs at the foot of the flank, slope 6–15°, chaotic composition; surface erosion | Golden red soil changes due to wet rice cultivation<br>Yellow soil on clay |
| Level 5 | Vacant land<br>Water surface | Chute drainage flow; V-shaped cross-sectional area, straight vertical rectilinear or hierarchical display of erosion terrace; deep erosion<br>Eroded trough—accumulated; V-shaped, narrow U-shaped cross; erosion—accumulation | Valley land due to condensation products<br>Yellow brown soil on old silt |

**Table 3.** Flash flood risk evaluation indicators for average rainfall of several years (RM), topographic wetness index (TWI), average slope of tributary (SB), landslide density (LS) and river density (RD) information layers.

| Evaluation Level | TWI | LS (Score/km$^2$) | SB (Degrees) | RM (mm) | RD (km/mm$^2$) |
|---|---|---|---|---|---|
| Level 1 | 0–0.9 | 0.01–0.02 | >40 | <150 | 0–0.5 |
| Level 2 | 0.9–1.3 | 0.02–0.04 | 30–40 | 150–250 | 0.5–1.5 |
| Level 3 | 1.3–3.2 | 0.04–0.06 | 20–30 | 250–350 | 1.5–2.5 |
| Level 4 | 3.2–5.4 | 0.06–0.08 | 10–20 | 0–450 | 2.5–3.5 |
| Level 5 | 5.4–15.4 | 0.08–0.1 | <10 | >450 | 3.5–4.5 |
| **Evaluation level** | **TWI** | **LS (score/km$^2$)** | **SB (degrees)** | **RM (mm)** | **RD (km/mm$^2$)** |
| Level 1 | 0–0.9 | 0.01–0.02 | >40 | <150 | 0–0.5 |
| Level 2 | 0.9–1.3 | 0.02–0.04 | 30–40 | 150–250 | 0.5–1.5 |
| Level 3 | 1.3–3.2 | 0.04–0.06 | 20–30 | 250–350 | 1.5–2.5 |
| Level 4 | 3.2–5.4 | 0.06–0.08 | 10–20 | 350–450 | 2.5–3.5 |
| Level 5 | 5.4–15.4 | 0.08–0.1 | <10 | >450 | 3.5–4.5 |

The results of the evaluation of information layers are shown in Figure 3.

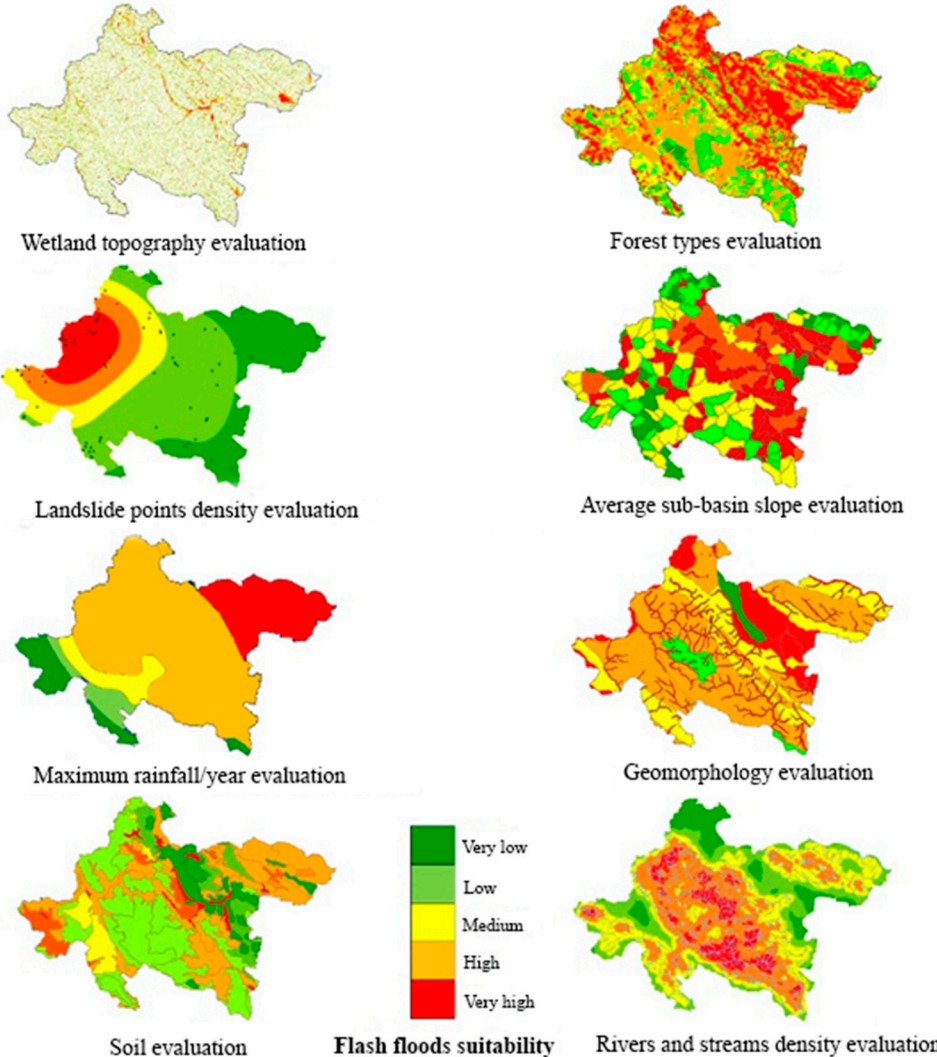

Wetland topography evaluation

Forest types evaluation

Landslide points density evaluation

Average sub-basin slope evaluation

Maximum rainfall/year evaluation

Geomorphology evaluation

Very low
Low
Medium
High
Very high

Soil evaluation

**Flash floods suitability**

Rivers and streams density evaluation

**Figure 3.** Evaluation of flood risk information layers in the Thuan Chau district.

*4.2. Calculation of Ratings for Information Layers using AHP*

The AHP method was adopted to calculate the ratings for each layer [10,18]. The results are shown in Table 4.

**Table 4.** Calculation matrix to determine the ratings of each information layer.

| Info Layer | TWI | FR | LS | SB | RM | GM | S | RD | Total | Ratings |
|---|---|---|---|---|---|---|---|---|---|---|
| TWI | 0.33 | 0.26 | 0.23 | 0.25 | 0.45 | 0.34 | 0.18 | 0.22 | 2.27 | 0.28 |
| FR | 0.11 | 0.09 | 0.14 | 0.17 | 0.06 | 0.06 | 0.16 | 0.13 | 0.91 | 0.11 |
| LS | 0.07 | 0.03 | 0.05 | 0.03 | 0.04 | 0.04 | 0.11 | 0.09 | 0.44 | 0.05 |
| SB | 0.11 | 0.04 | 0.14 | 0.08 | 0.08 | 0.06 | 0.13 | 0.13 | 0.77 | 0.11 |
| RM | 0.16 | 0.35 | 0.28 | 0.25 | 0.23 | 0.34 | 0.18 | 0.22 | 2.02 | 0.25 |
| GM | 0.11 | 0.18 | 0.14 | 0.17 | 0.08 | 0.11 | 0.11 | 0.13 | 1.02 | 0.13 |
| S | 0.05 | 0.01 | 0.01 | 0.02 | 0.03 | 0.03 | 0.03 | 0.01 | 0.19 | 0.02 |
| RD | 0.07 | 0.03 | 0.02 | 0.03 | 0.05 | 0.04 | 0.11 | 0.04 | 0.38 | 0.05 |
| | | | | | | | Consistency Index (CI) | | | 0.08 |
| | | | | | | Random Consistency Index by N (RI) | | | | 0.41 |
| | | | | | | | Consistency Ratio (CR) | | | 0.056 |

The CR reliability of the matrix is determined by the CR common index. If the value of the CR index is less than or equal to 0.1, then the consistency between the factors in the comparison matrix is guaranteed. For the eight factors included in this research, CR = CI/RI = 0.08/1.41 = 0.056 (an index value <0.1 indicates that reliability is warranted). The method for calculating the indicators is given in the documents of Saaty [20,21].

*4.3. Formulation of the Thuan Chau Flash Flood Risk Map Using the MCA Model*

The potential flash flood risk map is formulated by the following function [4,21,23]:

$$FFPI = \sum_{i=1}^{n} W_i X_i, \tag{2}$$

where *FFPI* is the potential flash flood risk, $W_i$ refers to ratings of the factor (i), $X_i$ refers to the factor (i) and n is the number of factors (1 to n-information layers mentioned in Section 3.3).

With the ratings calculated in Table 3, the function is specified as equivalent (Equation (3)):

$$\begin{aligned} FFPI = 0.28 * TWI + 0.11 * F + 0.05 * LS + 0.1 * SB + 0.25 * R + 0.13 * GM \\ + 0.02 * S + 0.05 * DS. \end{aligned} \tag{3}$$

The result obtained is the flash-flood risk map with different numerical values (Figure 4). In its original form, it has not yet featured an early warning map of a flash-flood risk.

By comparing the data and the locations of flash floods over the past few years, from 2000 to the present, there is a correlation between reality and forecast. In the Muoi stream basin of the Thuan Chau district, according to the statistics from 2000 to the present, a flash flood begins to arise when rainfall reaches 150 mm/d. At the same stream, there are also many places with a high risk of flash floods, which is explained by the influence of various parameters of the buffer.

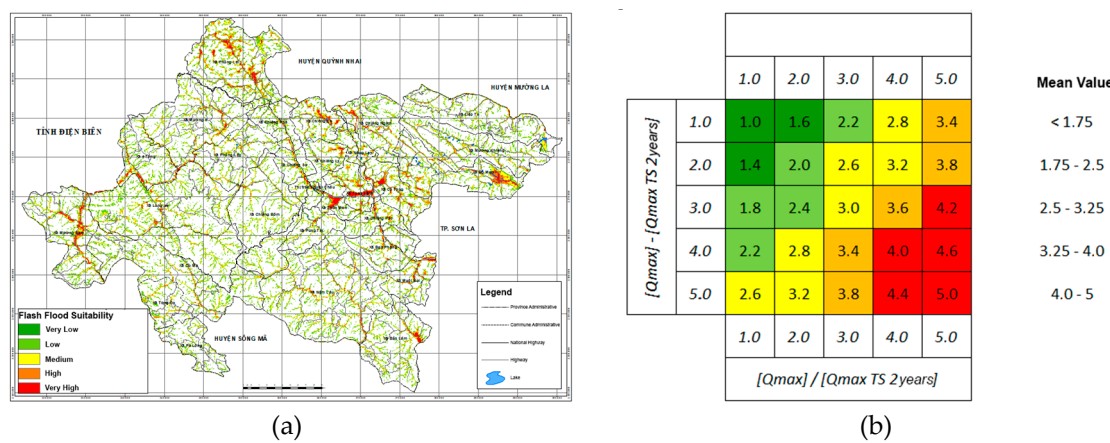

| | | 1.0 | 2.0 | 3.0 | 4.0 | 5.0 | | Mean Value |
|---|---|---|---|---|---|---|---|---|
| | 1.0 | 1.0 | 1.6 | 2.2 | 2.8 | 3.4 | | < 1.75 |
| | 2.0 | 1.4 | 2.0 | 2.6 | 3.2 | 3.8 | | 1.75 - 2.5 |
| [Qmax] - [Qmax TS 2years] | 3.0 | 1.8 | 2.4 | 3.0 | 3.6 | 4.2 | | 2.5 - 3.25 |
| | 4.0 | 2.2 | 2.8 | 3.4 | 4.0 | 4.6 | | 3.25 - 4.0 |
| | 5.0 | 2.6 | 3.2 | 3.8 | 4.4 | 5.0 | | 4.0 - 5 |
| | | 1.0 | 2.0 | 3.0 | 4.0 | 5.0 | | |
| | | | | [Qmax] / [Qmax TS 2years] | | | | |

(a)                                    (b)

**Figure 4.** (**a**) Flash flood risk map and (**b**) flash flood risk information (FFPI) for each pixel.

### 4.4. Inspection of the Accuracy of the Thuan Chau Flash-Flood Risk Map

Figure 5 shows the similarities between locations with a high risk of a flash flood (FFPI) on the risk map and locations where local flash floods occurred. The comparative results on the map show that flash-flood locations in the past were in high-risk areas. To test the accuracy of the research results, the receiver operating characteristic (ROC) method was used. In signal detection theory [23,24], the ratio of the received signal (accurate) to the total signal generated is used to evaluate the accuracy of the system. From the received signal, the Statistical Product and Services Solutions (SPSS) software can analyze and draw a signal acquisition characteristic curve (ROC) to determine the reliability of the system. Each point on the curve is the coordinates corresponding to the actual signal frequency on the vertical axis and the theoretical signal frequency on the horizontal axis (Figure 6).

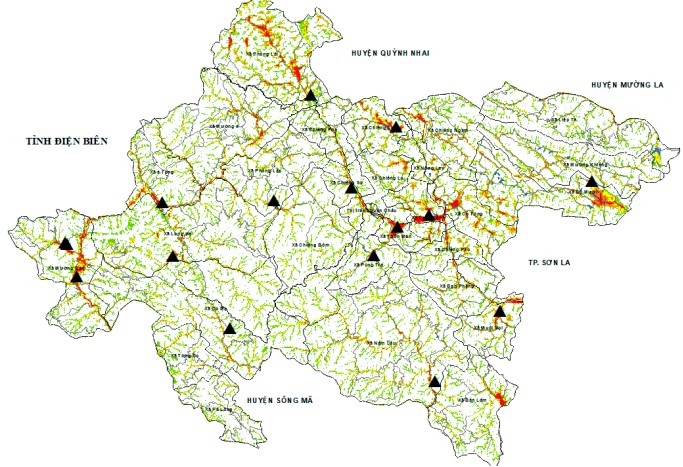

**Figure 5.** Comparison of forecast results with historical floods.

The higher the area under curve (AUC) value is, the better the distinction between the two states of precision (value 1) and inaccuracy (value 0). As the AUC increases, the curve approaches the upper horizontal axis with AUC = 1. The determination of accuracy is based on the following ratings: 0.80–0.90 = very good (A), 0.60–0.70 = good (B) and 0.50–0.60 = wrong (C). Comparing with past flash-flood data and the flash-flood risk map (Figure 6), the ROC curve in Figure 5 for AUC = 0.86 indicates that the accuracy is relatively good.

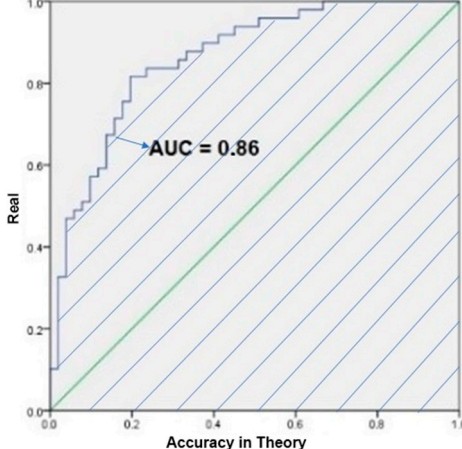

**Figure 6.** Receiver operating characteristic (ROC) curve constructed from the historical flash flood screening system compared with the Thuan Chau flood risk map. AUC: area under curve.

*4.5. Formulation of the Flash Flood Warning Map for Thuan Chau District*

At the present, there are many traditional meteorology stations to measure accumulative rainfall for the province. Additionally, AgriMedia (Hanoi, Vietnam) have set up more than 100 iMETOS stations, of which three iMETOS stations have been chosen for analyzing and setting the threshold. Therefore, we integrated these between auto-rainfall stations and traditional meteorology stations for analyzing.

The solar-powered automatic weather station system developed for model 2 consists of three iMETOS stations which are able to connect online in a "two-way" manner with the Meteoblue Global Meteorological Center of Switzerland. By applying Equation (3), with information layers of 1-day forecast (or two to six-day forecasts), the early warning map is established. In Thuan Chau district, based on the statistics from the 2000–2017 period and the evaluation model, the rain threshold for generating a flash flood (Rmax) was divided into five levels: Level 5 Rmax ≥ 300 mm/d; Level 4: 250 < Rmax < 300 mm/d; Level 3: 200 < Rmax < 250 mm/d; Level 2: 150 < Rmax < 200 mm/d; and Level 1: Rmax < 150 mm/d [22–24]. Figure 7A,B show the results of flash-flood forecast processing by rainfall in the province. In locations where rainfall exceeds the threshold, flash floods will occur (Figure 7c).

*4.6. Structure of Flash Flood Warning System*

The rainfall forecast information processing for each iMETOS weather station system [10,25,26] and integration with risk maps for flash flood early warning is done by webGIS. Accordingly, information is transferred to the website to provide flash flood warning information to users [15,16,22]. The generalized model and key features of the system are shown in Figure 8; this figure shows the completed resolutions for an early warning system for that study site. Input data, such as daily weather data (daily rainfall, daily temperature) and management documents, together with the hardware, have been analyzed, accessed, stored and displayed by disaster-management software. Using webGIS tools, information that is very useful for the decision will be displayed on the user interface. The residents can receive early-warning alerts via SMS message, streaming video and as electronic documents.

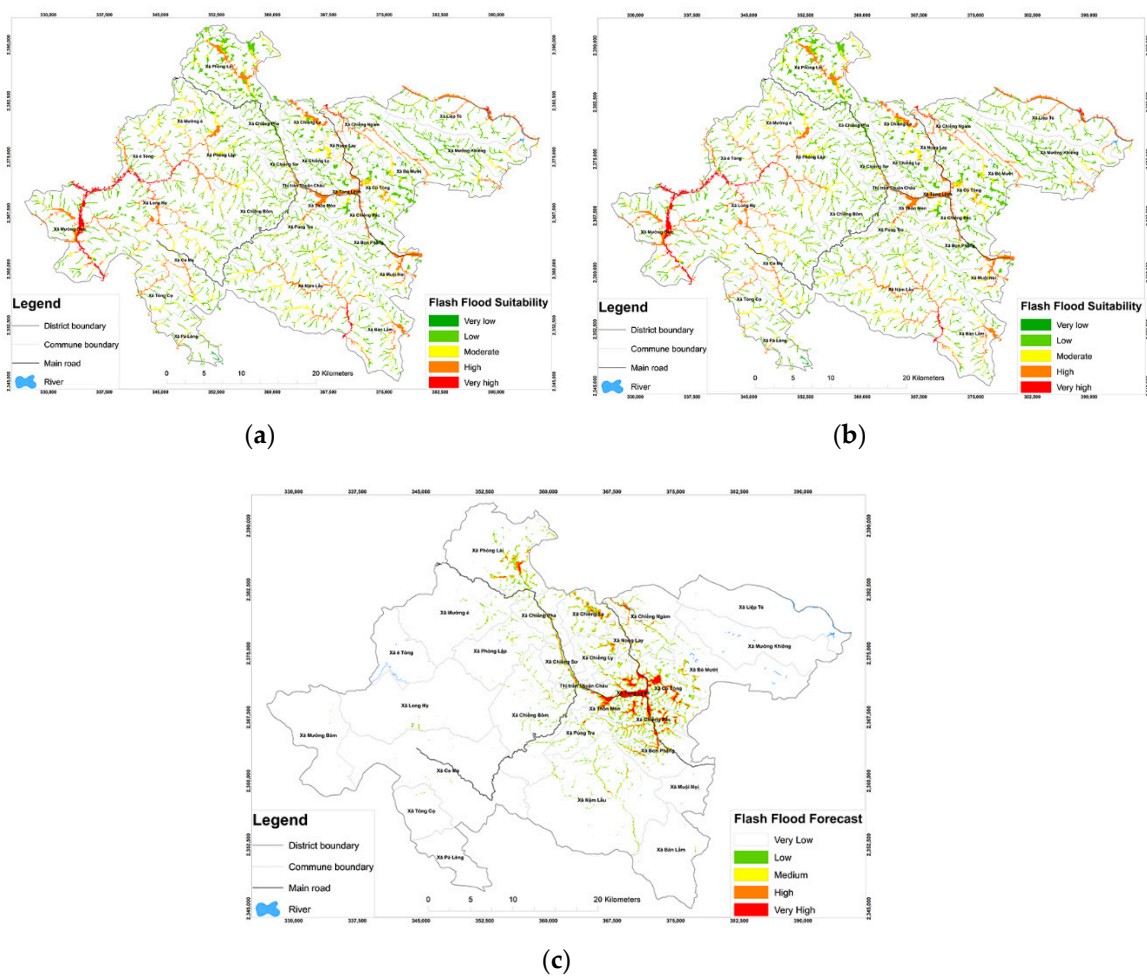

**Figure 7.** Results of flash-flood forecast processing by rainfall in Thuan Chau district: (**a**) Forecast rainfall map, (**b**) flash-flood risk map and (**c**) early flash-flood forecast map by the forecast rainfall, with a flash flood generation threshold of 150 mm/d [13].

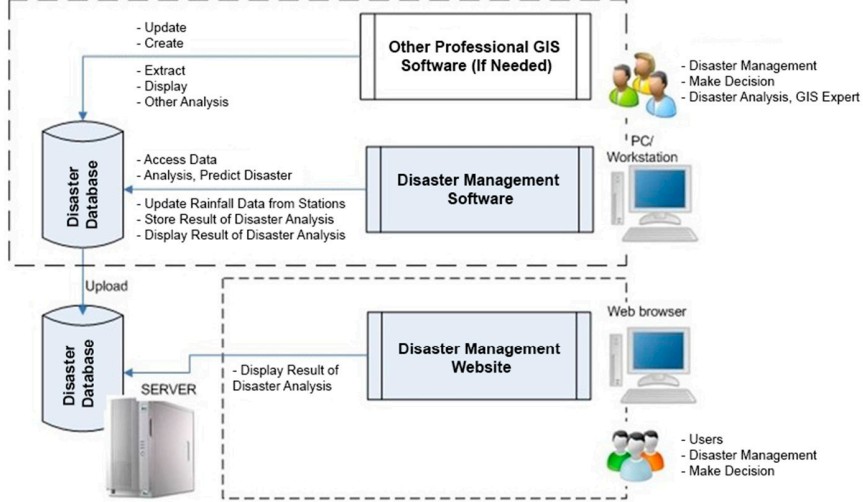

**Figure 8.** Generalized model and key features of flash flood warning software.

The software was developed using the open source programming language Python, Personal Home Page (PHP) and the PostgreSQL/PostGIS database [27]. The software was simply organized using the key functions: Environmental Systems Research Institute (ESRI)-standardized database

management, spatial interpolation of rainfall and integration of information layers to formulate a flash flood forecast map and the map was exported in the form of data and metrics. From the software, information is transferred to the website to provide information online and export messages to users.

## 5. Conclusions

With a shortage of or limited survey data, the application of GIS with the approach of accessing basins facilitates the formulation of a flash flood risk map for mountainous areas.

In natural disaster management, a flash flood warning system [28] can be a single integrated piece of software or can also include software and webpages operating in different phases (Figure 9a,b). However, they should meet three basic functions: quick processing of forecast rainfall data, rapid provision and communication about possible locations of disasters for concerned people in message form. The development of an early-warning system should follow both technology and application directions [29,30]. At present, the model has been tested and aligned to produce forecasts close to reality.

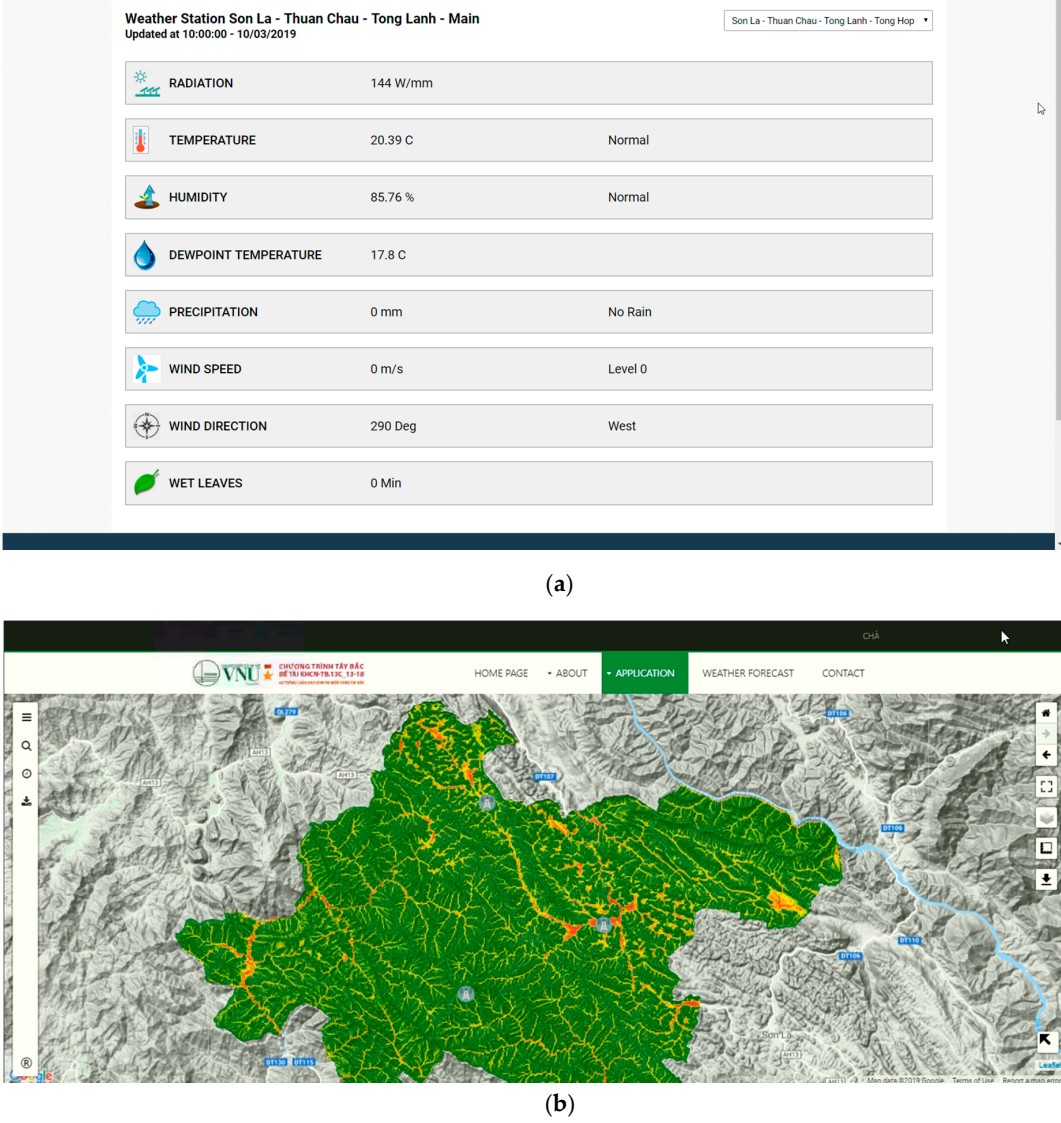

(**a**)

(**b**)

**Figure 9.** Main interface of the software (**a**) and the risk-warning website (**b**).

In this study, the basin parameters were processed using the MCA model, through which the entire basin area was evaluated by pixel and the results map is shown in detail, meeting the requirements of the district-level research scale and commune-level scale. The accuracy of theory is 0.86, which indicates that the accuracy is relatively good. The iMETOS automatic meteorological station system and webGIS are useful in decision-making regarding early warning flood alerts.

The system has been adopted for early warnings of flash floods in Thuan Chau district for disaster management in Son La Province. The research results have been transferred to the district and can possibly be expanded to other districts in northern mountainous area of Vietnam.

**Author Contributions:** Conceptualization, Ngoc Thach Nguyen and Tien Yin Chou; Data curation, Mei Ling Yeh and Quoc Huy Nguyen; Formal analysis, Yao Min Fang and Quoc Huy Nguyen; Funding acquisition, Tien Yin Chou, Yao Min Fang, Mei Ling Yeh and Thanh Van Hoang; Investigation, Thanh Van Hoang, Ngoc Thach Nguyen and Tien Yin Chou; Methodology, Ngoc Thach Nguyen and Mei Ling Yeh; Project administration, Ngoc Thach Nguyen; Resources, Ngoc Thach Nguyen and Xuan Linh Nguyen; Software, Yao Min Fang, Quoc Huy Nguyen and Xuan Linh Nguyen; Supervision, Tien Yin Chou and Yao Min Fang; Validation, Ngoc Thach Nguyen and Tien Yin Chou; Visualization, Quoc Huy Nguyen and Xuan Linh Nguyen; Writing – original draft, Thanh Van Hoang and Quoc Huy Nguyen; Writing—review and editing, Tien Yin Chou and Thanh Van Hoang.

**Funding:** This research received no external funding.

**Acknowledgments:** This article is the result of a state-level project titled "Research on modeling and system of sub-regional weather forecasting and warning of flood, forest fire and agricultural pests at district level in the North West Vietnam", Code: KHCN-TB.13C/13 -18 and has been financed by National Program for Tay Bac, VNU Hanoi, Viet Nam; supervised and assisted by Geographic Information Systems Research Center, Feng Chia University, Taiwan.

**Conflicts of Interest:** The authors declare no conflict of interest.

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
