# Peer review of "A Robust Early Warning System for Preventing Flash Floods in Mountainous Area in Vietnam"

_ijgi, doi:10.3390/ijgi8050228_

Round 1

Reviewer 1 Report

1. How to identify the evaluation criteria (Page 7,line 210)?

2. Why does not the multi-indicator analysis(MCA) take into account of the vulnerability of people and society? The risk of flash flood is commonly due to the hazards, exposure and vulnerability of disaster (from your reference 4).

3. What is the prediction period of early warning time? several Days or hours? The detailed explantion of the method and implement process should be given.

4. This paper use AHP to calculate the weight value of each indicator.How to get the  weight value of each indicator? Whom are you investigating with? How to reduce the subjectivity in this study

Author Response

Response to Reviewer 1

Point 1. How to identify the evaluation criteria Page 7line 210?

Response to Point 1: Dear Editor and Reviewer 1:

The evaluation criteria of flooding are identified by references first (25, 4). After that we also use knowledge of us and local experts to identify what is potential indicators in our study area for flooding.

Point 2. Why does not the multi-indicator analysis(MCA) take into account of the vulnerability of people and society The risk of flash flood is commonly due to the hazards, exposure and vulnerability of disaster (from your reference 4).

Response to Point 2: Dear Editor and Reviewer 1:

In this research, we only concentrate on natural indicators of flooding, people and society indicators aren’t mentioned. Because in our study area, flooding is not affected too much by people and society.

Point 3. What is the prediction period of early warning time? several Days or hours? The detailed explanation of the method and implement process should be given.

Response to Point 3: Dear Editor and Reviewer 1:

Our prediction period is 1 day (24 hours). The automation process is set up on system to run our forecast flooding model day by day and return results to display on WebGIS.

Point 4. This paper use AHP to calculate the weight value of each indicator. How to get the weight value of each indicator? Whom are you investigating with? How to reduce the subjectivity in this study?

Response to Point 4: Dear Editor and Reviewer 1:

We use two ways to get the weight of each indicator and reduce the subjectivity in this study:

- First, we base on past research (References 25, 4).

- Second, we base on knowledge from local expert. We create a survey from local experts who have many experiences in forecast flooding model and natural characteristics in our study area.

After that, we integrated them and calculated weights using AHP method as you can see the results in Table 4.  

We sincerely thank you so much for your comments and support us a lot. Best wish for you!

Sincerely yours,

On behalf of authors team:

Ms Van

Reviewer 2 Report

Dear authors,

I read the article and I recommend some improvements.

I hope my suggestions make the article better, and that it will be accepted.

Author Response

Response to Reviewer 2

A Robust Early Warning System for Preventing Flash Floods in Mountainous Area in Vietnam

Dear authors

I read carefully your article and found some errors of the article.

Point 1. Line 61 Here is the first citation in the article with the order number [26]. Being the first citation in the text, it will be numbered with [1], and the remaining citations need to be further counted with 2, 3, ... …

Response to Point 1: Dear Editor and Reviewer 2:

We have follow the MDPI format and MDPI english editing service twice for Format, so I think the references should be arranged from A-Z. That why we put the order number [26] because [26] is Website. Is that correct? Please feel free to let me know, I am sincerely thank you so much.

Point 2. 29 (29 citations are referenced). Please modify the number system of bibliographic titles quoted in the article as requested by this journal!

Response to Point 2: Dear Editor and Reviewer 2:

Yes, we have followed your suggestions and done (as highlight in the paper)

Point 3: Line 89 I recommend the replacement of the term solid articles with debris.

Be careful! The purpose of this research is not clearly stated in the introductory part.

Response to Point 3: Dear Editor and Reviewer 2

Yes, we have followed your suggestions and done (as highlight in the paper)

Point 4: Line 121 What is the importance of who lives in the studied area, I think other geographic features are more relevant to the hydrological forecast.

Nothing is said about land use in the introductory part.

Response to Point 4: Dear Editor and Reviewer 2:

Yes, we have followed your suggestions and done (as highlight in the paper)

Point 5: Line 123 Figure 1 - Mathematical coordinates are missing from the map.

Response to Point 5: Dear Editor and Reviewer 2:

Yes, we have followed your suggestions and done (as highlight in the paper)

Point 6: Line 141 Figure 2 We noticed the numbering of the figures with a and b, but the border of the text is exceeded on the right side of the image. Please rearrange the figure on the page.

Response to Point 6: Dear Editor and Reviewer 2:

Yes, we have followed your suggestions and rearrange the fugure already, now it can be seen the right side of image.

Point 7: Line 199 Fill in the text as follows: "         the model is (Eq.1):"

Response to Point 7: Dear Editor and Reviewer 2:

Yes, we have followed your suggestions and done (as highlight in the paper)

Point 8: Line 226      wi        xi !!!!!

Response to Point 8: Dear Editor and Reviewer 2:

Yes, we have followed your suggestions and done (as highlight in the paper)

Point 9: Line 228 To complete the text as follows: ".         as equivalent (Eq.3):"

Response to Point 9: Dear Editor and Reviewer 2:

Yes, we have followed your suggestions and done (as highlight in the paper)

Point 10: Caution: Figures 5 and 6 will be numbered from left to right (Figure 5 and Figure 6)

Response to Point 10: Dear Editor and Reviewer 2:

Yes, we have followed your suggestions and done (as highlight in the paper)

Point 11: Figure 9 Increase the resolution of the figures presented and switch to the English version in order for them to be understood by the general public.

Note: The figure is placed after it has been quoted in the text (Line 293 Figure 9 and line 298 citations in the text and this figure).

Response to Point 11: Dear Editor and Reviewer 2:

Yes, we have followed your suggestions and done (as highlight in the paper)

Point 12: In view of the conclusions presented, should it be mentioned in the introductory part the hydrological forecasts as an objective of the research?

Response to Point 11: Dear Editor and Reviewer 2:

Yes, we have followed your suggestions and write more in the conclusions part (as highlight in the paper)

Article Title A Robust Early Warning System for Flood Prevention in Vietnamese Mountains leads to the idea of hydrological forecasting.

Thank you for taking your previous suggestions into consideration. All the best,

Daniel

Dear the Editor and Reviewer,

On behalf of the authors team, I am sincerely thank you so much for your comments, your suggestions are really useful for me, I have learned a lot from this.

Best wish for you!

Sincerely yours,

Ms Van

This manuscript is a resubmission of an earlier submission. The following is a list of the peer review reports and author responses from that submission.

Round 1

Reviewer 1 Report

1. The literature review of this study is not presented in this paper. So, the innovation and contribution of this study cannot be judged clearly.

2. The early warning system of the title is not related to the content, which only refers to a framework of the early warning system. And how the early warning system operates, whether the data is available, and what the results of the early warning and the information provided to users are not clearly illustrated.

3. It is mentioned that IMETOS is used to predict the rainfall in the next 1-6 days. How is it combined with the flash flood risk assessment in the text of this paper? What is the relation between the early warning of flash flood and the prediction of rainfall? Whether this early warning system fail when the rainfall is inaccurate?

4. It mentions the lack of meteorological and hydrological stations in small and medium-sized river basins in Page 4. How to apply the prediction results of IMETOS rainfall to the early warning system? How is Maximum rainfall data spatially distributed? What methods are used and whether the accuracy meets the evaluation criteria?

5. It mentions "Where the rainfall exceeds the threshold level, it is possible to quickly identify and provide justifiable" in Page 3. Is the flash flood warning only related with the rainfall forecast?

6. The multi-indicator analysis (MCA) is mentioned in this paper. But how is the relationship between each indicator and flash flood disaster warning is not explained? The indicator system does not take into account the vulnerability of people and society.

7. Please declare the innovative part of this study compared with the existing research

9. In the table of Flash flood risk evaluation for geomorphology (GM), soil (S), forest (FR) information, there are too many qualitative analyses, which can not meet the precision requirement of early warning of flash flood.

10. Most of the content of this paper is about flash flood risk assessment, but multi-indicator analysis does not reflect the characteristics of flash flood in this district, and the risk assessment is static, which cannot support the early warning system of mountain flood.

Author Response

Response to Reviewer 1 Comments

Point 1: The literature review of this study is not presented in this paper. So, the innovation and contribution of this study cannot be judged clearly.

Response 1: Dear the Reviewer and Dear the Editor! Thank you very much for your comments. We have edited and cite some references for our papers in the part of 1.1 (purpose of this work and its background)

Point 2: The early warning system of the title is not related to the content (Change the title), which only refers to a framework of the early warning system. And how the early warning system operates, whether the data is available, and what the results of the early warning and the information provided to users are not clearly illustrated.

Response 2:

Dear the Reviewer and Dear the Editor: We have changed the title to meet the content

We have shown it on the Figure 2: The rainfall amount from IMETOS station will send back all real-time information through the internet, ADSL, 4G, the data will be analysed at the Emergency Operation Center, then the Emergency Operation Center will send the early alert to the National Flood Warning Center, warning via mobile and computer.

Whenever, wherever they are, the results of the early warning and the information provided to users the early alert information about high potential of Flash floods area thought the internet and mobile devices. Accordingly, the local people and managers can make appropriate decisions to prevent natural disasters

Point 3: It is mentioned that IMETOS is used to predict the rainfall in the next 1-6 days. How is it combined with the flash flood risk assessment in the text of this paper? What is the relation between the early warning of flash flood and the prediction of rainfall? Whether this early warning system fail when the rainfall is inaccurate?

Response 3: Dear the Reviewer and Dear the Editor:

It is mentioned that IMETOS is used to predict the rainfall in the next 1-6 days. How is it combined with the flash flood risk assessment in the text of this paper? (response: Firstly, we developing a potential flood risk map; Secondly: We setting up an automatic climate system that can be connected in two directions with the Global meteorological network; and thirdly: building WebGIS software online in the Internet to quickly process rainfall forecasts.

Compared with the data and location of flash flood events in the past since 2000, there is a good correlation between reality and forecast. At the same stream, there are also many places where there is a risk of flash flooding, which can be explained by the influence of various parameters of the catchment basin surface.

What is the relation between the early warning of flash flood and the prediction of rainfall? (response: The precipitation can be quickly identified when it exceeds the threshold, and the information about the flood including time and position will be timely provided in different levels)

Whether this early warning system fail when the rainfall is inaccurate? (response: We did validate and calibrate for model, the accuracy (Figure 6) shown that AUC = 85%, that is good, it can not be failed)

Point 4: It mentions the lack of meteorological and hydrological stations in small and medium-sized river basins in Page 4. How to apply the prediction results of IMETOS rainfall to the early warning system? How is Maximum rainfall data spatially distributed? What methods are used and whether the accuracy meets the evaluation criteria?

Response 4: Dear the Reviewer and Dear the Editor

It mentions the lack of meteorological and hydrological stations in small and medium-sized river basins in Page 4. How to apply the prediction results of IMETOS rainfall to the early warning system?

Response: The rainfall forecast information processing for each IMETOS weather station system and integration with risk charts for flash flood early warning is done by WebGIS. Accordingly, information is transferred to the website to provide flash flood warning information to users. The generalized model and key features of the system are shown in Figure 8.

How is Maximum rainfall data spatially distributed? What methods are used and whether the accuracy meets the evaluation criteria?

Response: We design the software that was developed on the open source programming language Python, PHP, and the PostgreSQL/PostGIS database . The software was simply organized with the key functions: ESRI-standardized database management, spatial interpolation of rainfall (using Kriging method), and integration of information layers to formulate a flash flood forecast chart, and the chart was exported in form of data and metrics.

From the software, information is transferred to the website to provide information online and export messages to users.

The methods are validated with AUC is 86% so it can be met the evaluation criteria.

Point 5: It mentions "Where the rainfall exceeds the threshold level, it is possible to quickly identify and provide justifiable" in Page 3. Is the flash flood warning only related with the rainfall forecast?

Response 5: Dear the Reviewer and Dear the Editor!

The flash flood warning is not only related with the rainfall forecast, but also need to based on those maps that are shown in Figure 3 (Wetland topography, forest evaluation, landslide points density evaluation, average sub-basin slope evaluation, Maximum rainfall/year evaluation, geomorphology evaluation, Soil evaluation and River and streams evaluation map)

Point 6: The multi-indicator analysis (MCA) is mentioned in this paper. But how is the relationship between each indicator and flash flood disaster warning is not explained? The indicator system does not take into account the vulnerability of people and society.

Response 6: Dear the Reviewer and Dear the Editor!

The study focused on identifying the baseline flooding factors, including: soil properties, vegetation cover, basin slope, river density, cumulative flow; and comparing with statistical data to classify the possibility of flash floods of each mapping unit. The flash floods will form when precipitation exceeds the threshold and the risk of flash floods will be determined throughout the basin. GIS application will determine the weight, and then make the integration between the factors with their respective weight, and graphing of flow charts in individual grid cells.

This research only focus on qualitative and will analyse the vulnerability and society in the future research.

Point 7: Please declare the innovative part of this study compared with the existing research?

Response 7: Dear the Reviewer and Dear the Editor!

Flash floods are the type of large flood which is suddenly occur in very short time in small basins with steep terrain and high velocity that are very common in Viet Nam. Thus, flash floods have the significant destruction as compared with normal floods. The formation of flash flood is closely related to the rainfall intensity, climatic condition, terrain properties, human activities and flood drainage condition.

In addition, we set up IMETOS the development of an IMETOS automatic meteorological station system. These stations should have a 10–15km active radius, be directly connected to the global meteorological network (www.meteoblue.com), and receive any information about meteorological conditions in the past 30 days, the current weather, and weather forecasts for 1–6 days for the station location.

Comparing those existing research which are not suitable for the mountainous area in Vietnam.

In this study, authors are using MCA (Multi-Indicator Model) and FFPI (Flash Flood potential Index) are suitable to the mountainous area in Vietnam.

Point 8: In the table of Flash flood risk evaluation for geomorphology (GM), soil (S), forest (FR) information, there are too many qualitative analyses, which can not meet the precision requirement of early warning of flash flood.

Response 8: Dear the Reviewer and Dear the Editor!

The study focused on identifying the baseline flooding factors, including: soil properties, vegetation cover, basin slope, river density, cumulative flow; and comparing with statistical data to classify the possibility of flash floods of each mapping unit. The flash floods will form when precipitation exceeds the threshold and the risk of flash floods will be determined throughout the basin. GIS application will determine the weight, and then make the integration between the factors with their respective weight, and graphing of flow charts in individual grid cells.

The system diagram of the multi-indicator process is shown in the following diagram:

Point 9: Most of the content of this paper is about flash flood risk assessment, but multi-indicator analysis does not reflect the characteristics of flash flood in this district, and the risk assessment is static, which cannot support the early warning system of mountain flood.

Response 9: Dear the Reviewer and Dear the Editor!

Upon comparing with the data and location of flash flood over the past few years, from 2000 to the present, there is a match between reality and forecast. In the Muoi stream basin of the Thuan Chau district, according to the statistics from 2000 to the present, a flash flood begins to arise when rainfall reaches 150 mm/d. At the same stream, there are also many places with a high risk of flash floods, which is explained by the influence of various parameters (multi-indicator) of the buffer.

So the real-time information and the static which support the early warning system of mountain flood.

Reviewer 2 Report

Dear authors

I read carefully your article and found some structural shortcomings of the article. Please read the file attached.

Author Response

Response to Reviewer 2 Comments

Point 1: Dear authors

I read carefully your article and found some structural shortcomings of the article.

Response 1: Dear the Editor and the Reviewer!

I am sincerely thank you very much for your helpful comments.

Point 2: The current introduction of the article is more in the study area, and it should be positioned after the introductory part and the methodology used in it.

Response 2: Dear the Editor and the Reviewer!

I have edited follow your suggestion, we have put that part as The study site (2.1 Study site) .

Point 3: The introductory part should contain the scientific context in which the research is carried out and the objectives to be addressed in the paper.

Response 3: Dear the Editor and the Reviewer!

We have re-write and add more information for scientific context, and show the objective to be addressed at the part 1.2 (1.2 Purpose of this work and its background)

Point 4: Chapter 2 can be the introductory part of the article, of course supplemented by several references to scientific development in this field of study. I would like this part to be restructured according to the scientific methodology of the magazine.

Response 4: Dear the Editor and the Reviewer!

We have edited as your suggestion to move a part of Chapter 2 to Introduction (Part 1) according to the scientific methodology of the journal.

Point 5: Line 40 Figure 1 - Marking the North direction and more legible writing on cartographic material.

Response 5: Dear the Editor and the Reviewer! Yes, we have added the North direction follow your comments.

Point 6: Line 133 Formula will be written in characters and in normal font size

Response 6: Dear the Editor and the Reviewer!

We have edited under your guide. Thank you J

Point 7: Line 172 Wi will be replaced with Wi and Xi with Xi

Response 7: Dear the Editor and the Reviewer!

We have edited under your guide. Thank you J

Point 8: Line Figure 5 - I recommend a more readable writing on the presented cartographic materials (generally)

Response 8: Dear the Editor and the Reviewer!

Under your guide, we have explain more for that cartographic material: As the AUC increases, the curve approaches the upper horizontal axis with AUC = 1. The determination of accuracy is based on the following ratings: 0.80–0.90 = very good (A), 0.60–0.70 = good (B), and 0.50–0.60 = wrong (C). Comparing with past flash flood data and the flash flood risk chart (Figure 5), the ROC curve formulated in Figure 6 for AUC = 0.86 indicates that the accuracy is relatively good

Point 9: Line 207 (Figure 7c) will be replaced by (Figure 7C), as per the inscription on cartographic material. I want you to change the size of the writing from the pictures presented to be read easily.

Response 9: Dear the Editor and the Reviewer!

Yes, We have changed it to the size of writing J

Point 10: Line 217 and 218 will be deleted

Response 10: Dear the Editor and the Reviewer!

Yes, we have deleted it J

Point 11: Figure 9 Image Restructuring

Response 11: Dear the Editor and the Reviewer!

We did restructure it already, thank you J

Point 12: I recommend you also mention when the web pages referenced were accessed.

Response 12: Dear the Editor and the Reviewer!

Point 13: I hope that my comments as well as the editing instructions for this article (https://www.mdpi.com/journal/ijgi/instructions#preparation) help you in drafting a good article.

Response 1: Dear the Editor and the Reviewer!

Thank you so much for your comments, we did take a close look MPDI instruction preparation and re-structure for our paper.

Reviewer 3 Report

The authors of the paper present an Early Warning System for Flash Floods In a Mountainous Area, of Viet Nam.

The topic is interesting, but the structure of the paper is very poor and should to totally revised: in the following some suggestions.

Introduction: this is not a real introduction but a simple description of the studied area. The introduction should describe the conceptual framework of the presented study.

Chapters 2.1 and 2,2: these chapters could be considered part of a possible introduction because they describe (shortly) some theoretical points that are considered in the used approach.

Chapter 2.3 can not be considered a chapter but a simple paragraph.

Chapter 2.4 could be considered a good incipit for the description of the proposed workflow — just an incipit.

Chapter 3: the most important point for an early warning is the definition of the threshold. All the other results are a corollary of the thresholds. To define a threshold, authors should introduce which dataset they decided to use, the approach (e.g. Intensity/duration) and the adopted algorithm. This should be described before the description of the gis method.

Line 119: these threshold values are local or general?

Line 123: this subdivision should have been published somewhere, a citation is mandatory

Chapter 3.3: the description of used data is eight lines long and a table. This can not be considered acceptable for a scientific publication.

Chapter 4.1, line 148: what does this sentence mean: “In reference to some published research [22,4], it is possible to evaluate the layers of basin surface associated with the flash flood risks in the following table (Tables 2 and 3).”?

Figure 4: the map presented in figure 4 should be feasible, but the process used to reach this point is not adequately described.

Chapters 4.4, 4.5, and 4.6 are the only interesting part of the paper. The description of these chapters, even if they are good, is very limited.

Discussion: this is not a discussion but a description of the studied area

Author Response

Response to Review 3 Comments

The authors of the paper present an Early Warning System for Flash Floods In a Mountainous Area, of Viet Nam.

The topic is interesting, but the structure of the paper is very poor and should to totally revised: in the following some suggestions.

Point 1: Introduction: this is not a real introduction but a simple description of the studied area. The introduction should describe the conceptual framework of the presented study.

Response 1: Dear the Reviewer and Dear the Editor!

We did edit and write more follow your suggestions.

Point 2: Chapters 2.1 and 2,2: these chapters could be considered part of a possible introduction because they describe (shortly) some theoretical points that are considered in the used approach.

Response 2: Dear the Reviewer and Dear the Editor!

Yes, your suggestions are really helpful for me, we change that part to the Introduction part.

Point 3: can not be considered a chapter but a simple paragraph.

Response 3: Dear the Reviewer and Dear the Editor!

Yes, we have edited that follow your comments

Point 4: Chapter 2.4 could be considered a good incipit for the description of the proposed workflow — just an incipit.

Response 4: Dear the Reviewer and Dear the Editor! Thank you

Point 5: Chapter 3: the most important point for an early warning is the definition of the threshold. All the other results are a corollary of the thresholds. To define a threshold, authors should introduce which dataset they decided to use, the approach (e.g. Intensity/duration) and the adopted algorithm. This should be described before the description of the gis method.

Response 5: Dear the Reviewer and Dear the Editor!

Yes, we have changed it follow your suggestions. Thank you J

Point 6: Line 119: these threshold values are local or general?

Response 6: Dear the Reviewer and Dear the Editor!

These threshold values are presented for Local government (Son La province, Viet Nam)

Point 7: Line 123: this subdivision should have been published somewhere, a citation is mandatory

Response 7: Dear the Reviewer and Dear the Editor! Yes, we have cited this J

Point 8: Chapter 3.3: the description of used data is eight lines long and a table. This can not be considered acceptable for a scientific publication.

Response 8: Dear the Reviewer and Dear the Editor!

We did explain and write more for this in the paper, thank you for your comments.

Point 9: Chapter 4.1, line 148: what does this sentence mean: “In reference to some published research [22,4], it is possible to evaluate the layers of basin surface associated with the flash flood risks in the following table (Tables 2 and 3).”?

Response 9: Dear the Reviewer and Dear the Editor!

That sentence we would like to explain more: the authors have referenced the previous researches, so it is possible to evaluate the layers of basin surface which related to the flash floods risk that we have shown in the Table 2.

Point 10: Figure 4: the map presented in figure 4 should be feasible, but the process used to reach this point is not adequately described.

Point 11: Chapters 4.4, 4.5, and 4.6 are the only interesting part of the paper. The description of these chapters, even if they are good, is very limited.

Response 11: Dear the Reviewer and Dear the Editor! Thanks so much for your comments, it is really useful for me to edit.

Point 12: Discussion: this is not a discussion but a description of the studied area

Response 12: Dear the Reviewer and Dear the Editor! Yes, we have edited that, thank you.

Round 2

Reviewer 1 Report

article has serious flaws, additional experiments needed, research not conducted correctly

Reviewer 2 Report

Dear authors,

I reread your article. It is well made and I appreciate the effort.

However, more corrections are needed to reach the publishing stage.

Here are some of the mistakes I found:

Figure 1. Left-side figure does not have a legend.

Figure 2 Suggests that the images are positioned one below the other, numbered with a and b.

Figure 3. A chart of the map representation would be required.

Figure 5 and 6. I recommend that they are repositioned on the page.

Figure 7. You should assign numbers to images A, B, C.

Figure 9. Choose another view to be easy to read.

Line 201: FFPI is the abbreviation of a Flash Flood Potential Index (FFPI), it's probably a translation error, please check it in the entire article.

Line 209 Delete this row.

Line 213 This line, being after the table should be left blank.

Line 228 Please, check the index. it should be written as wi and xi.

Line 232 This line should be left blank.

Line 287 This line should be deleted.

The FFPI map looks a lot like the Map Topography map (TWI).

The term TWI means Topographical Wetness Index, not Wetland topography evaluation. 

Reviewer 3 Report

The paper has not been significantly improved.

in the following some comments

Chapter 1.1 it is impossible a chapter like this one without any citations.

Figure 2: I think that the right part of the figure represents the workflow of the processing server mentioned in the left part of the picture. If yes, this should be better explained in the picture and in the caption

Chapter 3. In one page authors described: the dataset, the methodology, and Formulation of Space Model in Early Flash Flood Warning. I think that it cannot be considered sufficient for a high-level international journal.

Chapter 3.2: this is the key point of the paper. The description cannot be considered acceptable.

Chapter 3.3: the description of thresholds is too limited and not scientifically based

Chapter 4.5 the description of this chapter is very poor.

Chapter 4.6: “The rainfall forecast information processing for each IMETOS weather station system [23,24,25] and integration with risk charts for flash flood early warning is done by webGIS” what does it means? How is it performed? Again, this is a simple description of what happened, without a real description required for a scientific article.

ISPRS Int. J. Geo-Inf. EISSN 2220-9964 Published by MDPI AG, Basel, Switzerland RSS E-Mail Table of Contents Alert
Back to Top